# FROM MOTION TO BEHAVIOR: HIERARCHICAL MODELING OF HUMANOID GENERATIVE BEHAVIOR CONTROL

## ABSTRACT

Human motion generative modeling aims to synthesize complex motions from daily activities. However, current research is fragmented, focusing on either low-level, short-horizon motions or high-level, disembodied action planning, thereby neglecting the hierarchical and goal-oriented nature of human activities. This work shifts the research focus from motion generation to the more holistic task of **humanoid behavior modeling**. To formally address this, we first introduce **Generative Behavior Control (GBC)**, a new task focused on generating long-term, physically plausible, and semantically coherent behaviors from high-level intentions. To tackle this task, we present a novel framework that aligns motion synthesis with hierarchical plans generated by large language models (LLMs), leveraging principles from task and motion planning. Concurrently, to overcome the limitations of existing benchmarks, we introduce the **GBC-100K** dataset, a large-scale corpus annotated with hierarchical semantic and motion plans. Experimental results demonstrate our framework, trained on GBC-100K, generates more diverse and purposeful human behaviors with up to $10\times$ longer horizons than existing methods. This work lays a foundation for future research in behavior-centric modeling, with all code and data to be made publicly available.

## 1 INTRODUCTION

A key ambition in robotics and AI is to build humanoid agents capable of learning and executing complex skills from high-level human instructions. The human-like form factor of these agents presents a unique opportunity to leverage vast amounts of human data for operating in human-centric environments. Yet, progress is hampered by a fundamental fragmentation in current research: **motion generation** methods produce realistic but short-sighted movements (Zhang et al., 2025b; Dai et al., 2025; Feng et al., 2024a; Zhang et al., 2022; Lucas et al., 2022) lacking purpose (Liu et al., 2024; Shafir et al., 2024; Guo et al., 2022b); **physics-informed control** ensures stability (Yuan et al., 2023; Luo et al., 2023a;b; Tevet et al., 2024; Yan et al., 2024; Yao et al., 2024; He et al., 2024a) but remains disconnected from semantic goals (Truong et al., 2024; Serifi et al., 2024; Tessler et al., 2024); and **Task and Motion Planning (TAMP)** (Lin et al., 2024; Ortiz-Haro, 2024; Leung et al., 2024; Cheng et al., 2023; Garrett et al., 2021) is often too rigid and deterministic for the diversity of human behavior (Zhao et al., 2024). This reveals a critical gap: existing methods focus on the "how" of movement, not the "why" of behavior.

To address this fragmentation, we argue for a shift from motion generation to behavior generation. We formalize this by proposing and defining a new task: **Generative Behavior Control (GBC)**. The core challenge of GBC is to generate long-horizon action sequences that are simultaneously **(1) Goal-oriented**, **(2) Physically Plausible**, and **(3) Semantically Coherent**. Unlike mere motion synthesis (Lu et al., 2023; Zhang et al., 2022), GBC requires an agent to be **goal-oriented**, where it can decompose ambiguous, high-level instructions from an LLM (Wang et al., 2024; Brohan et al., 2023; Ahn et al., 2022; Huang et al., 2022b;

Chen et al., 2024b; Ding et al., 2023) into an executable, structured action plan. This task necessitates a unified approach that integrates high-level reasoning with low-level motor control, a challenge that existing benchmarks and methods are not designed to address.

Core to our solution for the GBC task are two synergistic contributions. The first is **the PHYLOMAN framework**, a novel architecture that satisfies the demands of GBC through a hierarchical synergy of LLM-based planning and physics-informed control. Its key innovation in motion synthesis is our proposed **Multi-segment Parallel Motion Diffusion Model (MP-MDM)**. MP-MDM follows a decoupled process, first determining keyframe poses and then interpolating transitions. We further co-design the transition and the target pose through a joint training scheme. This novel approach yields a more natural and kinematically coherent motion prior. Crucially, its "parallel-in-time" generation scheme allows for the **highly efficient synthesis of ultra-long sequences**, fully leveraging GPU capabilities to tackle generation horizons that were previously intractable. Our second contribution is the **GBC-100K** dataset, the first large-scale benchmark for behavior generation, whose multi-level textual annotations are crucial for learning the mapping from high-level goals to low-level actions, as shown in Table 1.

The synergy between our framework and dataset enables the **efficient and robust generation** of complex, whole-body behaviors, like tying a shoelace, standing up, and walking, that are $10\times$ **longer** than those from prior methods. Our contributions are thus three-fold, centered around the establishment and solution of our proposed task: (1) We introduce GBC-100K, a large-scale, hierarchically annotated dataset designed to support and evaluate the GBC task. (2) We propose PHYLOMAN, a hierarchical framework that provides the first effective and unified solution to the GBC task by integrating LLM-planning, our novel MP-MDM for generative modeling, and physics-based control. (3) We conduct extensive evaluations on both GBC-100K and HumanML3D (Guo et al., 2022a), which demonstrate that our PHYLOMAN significantly outperforms existing methods in generating long-horizon behaviors that are physically consistent and semantically faithful to high-level goals.

## 2 PRELIMINARY

### 2.1 GENERATIVE BEHAVIOR CONTROL

Generative Behavior Control (GBC) synthesizes long-term humanoid behaviors under both physical constraints and high-level semantic objectives. The task requires generating continuous, multi-minute motion sequences that maintain physical feasibility (e.g., joint limits, avoiding skating, and respecting contacts) and align with high-level instructions. Unlike traditional motion generation (Lu et al., 2023; Zhang et al., 2022), which addresses short-term dynamics, GBC focuses on the intentionality and semantic coherence of human behavior over long durations. It bridges the gap between low-level motor actions and overarching goals by formalizing the structure of goal-oriented behavior. In GBC, behavior is formally defined by a hierarchical script. A BehaviorScript $\mathcal{B} = \{\mathcal{D}, \mathcal{P}, \mathcal{A}\}$ consists of a high-level description $\mathcal{D}$, a set of PoseScripts $\mathcal{P} = \{p_0, \ldots, p_n\}$, and a set of MotionScripts $\mathcal{A} = \{a_0, \ldots, a_{n-1}\}$. The description $\mathcal{D}$ is an abstract summary of the sequence, following a structured template: [Subject], [Emotion/State/Style], [Action], [Direction/Goal], and [Environment/Background] (e.g., "a person **energetically** dancing in circles at a party"). Each PoseScript $p_i$ defines an atomic action (e.g., "raise right arm"), while each MotionScript $a_i$ captures the transition between poses $p_i$ and $p_{i+1}$, forming the complete sequence $(p_0, a_0, p_1, \ldots, a_{n-1}, p_n)$. This hierarchy formally defines the task's **goal-oriented** nature and the meaning of **behavior control**: the high-level description $\mathcal{D}$ acts as the semantic **goal**, realized through an executable plan of lower-level scripts. **Behavior control**, in turn, is the challenge of ensuring the synthesized motion faithfully executes this entire hierarchical plan. For example, a BehaviorScript with $\mathcal{D}$ describing "an excited person dancing" might decompose into MotionScripts like "spin energetically" and PoseScripts like "hold an upbeat pose," where each component inherits semantics from the abstract description. See Supp. A for details.

Table 1: **Comparison of Existing Motion-Language Benchmarks.** We comprehensively compare our proposed GBC-100K with widely adopted human generation benchmarks across multiple dimensions (list in columns left to right): the total number of human action sequences (#Seq.), whether is based on SMPL-parameterized model (Pavlakos et al., 2019) (SMPL) or/and video frames (Video), the total length of all videos (Len.), incorporation of hierarchical textual motion descriptions (Hierarchical), the number of distinct n-gram of words (Distinct-n@1) and phrases (Distinct-n@2) (Li et al., 2015), average length of the texts(Avg. Len.), whether with goal-oriented (Goal orient.) textual annotations, whether can support open-vocabulary (Open Vocab.) motion synthesis. Our proposed GBC excels at long-horizon, fine-grained descriptions and diverse outputs, leveraging a ∼100k-scale SMPL dataset with multi-level textual annotations.

| Datasets | #Seq. | SMPL | Video | Len. | Textual Annotations | | | | Goal Orient. | Open Vocab. |
|---|---|---|---|---|---|---|---|---|---|---|
| | | | | | Hierarchical | Distinct-n@1 ↑ | Distinct-n@2 ↑ | Avg. Len. ↑ | | |
| KIT-ML (Plappert et al., 2016) | 3.9K | ✓ | ✗ | 10.3h | ✗ | 0.88 | 0.86 | 8.43 | ✓ | ✓ |
| UESTC (Ji et al., 2019) | 25.6K | ✗ | ✓ | 83h | ✗ | 0.71 | 0.90 | 2.58 | ✗ | ✗ |
| NTU-RGB+D (Shahroudy et al., 2016) | 114.4K | ✗ | ✓ | 74h | ✗ | 0.69 | 0.88 | 3.12 | ✗ | ✗ |
| HumanAct12 (Guo et al., 2020) | 1.2K | ✓ | ✗ | 6h | ✗ | 0.73 | 0.89 | 1.97 | ✗ | ✗ |
| BABEL (Punnakkal et al., 2021) | - | ✓ | ✓ | 43.5h | ✗ | **0.90** | 0.81 | 1.43 | ✗ | ✗ |
| HumanML3D (Guo et al., 2022a) | 14.6K | ✓ | ✗ | 28.5h | ✗ | 0.46 | 0.86 | 12.37 | ✗ | ✓ |
| HMDB51 (Kuehne et al., 2011) | 6.8K | ✓ | ✓ | 7.8h | ✗ | 0.89 | 0.80 | 1.29 | ✗ | ✗ |
| COIN (Tang et al., 2019) | 46.3K | ✗ | ✓ | 476h | ✓ | 0.56 | 0.87 | 4.92 | ✗ | ✓ |
| ActivityNet (Caba Heilbron et al., 2015) | 2K | ✗ | ✓ | **648h** | ✓ | 0.33 | 0.76 | 13.48 | ✓ | ✓ |
| **GBC-100k** | **123.7K** | ✓ | ✓ | 250h | ✓ | 0.51 | **0.91** | **50.92** | ✓ | ✓ |

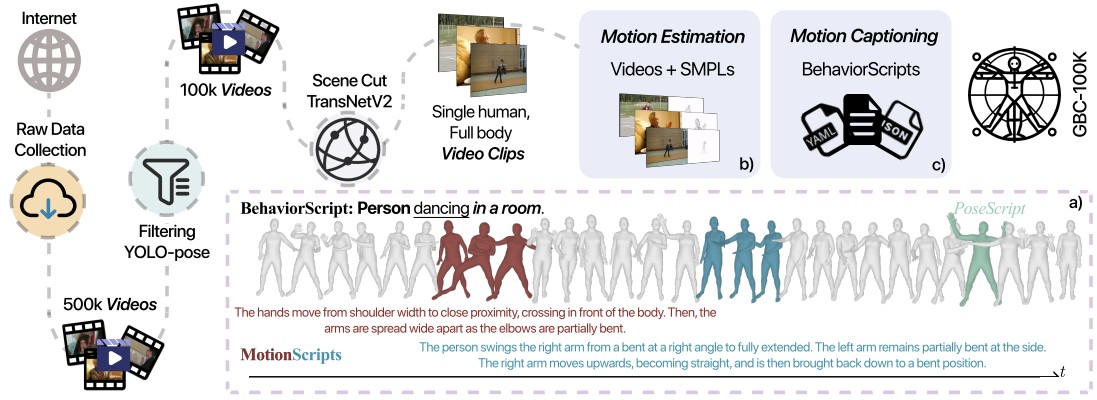

Figure 1: **Overview of the GBC-100K collection process.** We collect raw videos, filter for single full-body clips, then apply motion estimation and captioning to extract motions and annotations (a). Data are organized hierarchically (b) with behavior scripts, PoseScripts/motions, and SMPL sequences.

## 2.2 PROBLEM FORMULATION

We build our framework upon Task and Motion Planning (TAMP) (Garrett et al., 2021), a hybrid planning paradigm that integrates the discrete decision-making of task planning with the continuous constraints of motion planning to generate feasible action sequences for robots operating in complex environments. Task planning involves reasoning over symbolic variables, such as selecting optimal movement sequences, coordinating multi-step locomotion, or generating physical gestures to convey intent. Motion planning, on the other hand, focuses on computing collision-free paths and physically valid trajectories in the robot's configuration space. The seamless interaction between these two levels is essential to bridge the gap between abstract planning and physical execution.

Formally, the TAMP problem can be represented as finding a sequence $(x_0, a_0, x_1, a_1, \ldots, x_T)$, where $x_i \in \mathcal{X}$ are robot configurations and $a_i \in \mathcal{A}$ actions. Each action must satisfy task constraints $f(x, a) = \text{True}$

Figure 2: **Overview of the PHYLOMAN framework.** PHYLOMAN synthesizes behaviors via *task planning* (LLM → BehaviorScript) and *motion planning* (Diffusion-guided Control Policy → Humanoid action sequence), achieving goal-oriented, long-term humanoid behavior generation. The input behavior descriptions are analyzed by a Behavior Planning Network (*i.e.*, LLMs) to produce BehaviorScripts. We then project them into humanoid action space using motion diffusers, thereby guiding the Motion Tracking Policy to control the simulated humanoids.

and motion feasibility constraints $g(x_i, x_{i+1}) \leq 0$. The task planner operates in a symbolic space, often expressed using formal languages such as PDDL (McDermott et al., 1998), determining which action sequence achieves a given goal. We formulate Generative Behavior Control (GBC) as a hierarchical planning problem that transforms high-level language instructions into physically executable SMPL poses. The problem consists of two hierarchical levels: task-level planning and motion-level planning. At the task level, given an input prompt, the planner generates a sequence of PoseScripts $\{p_i\}_{i=0}^{n}$ and MotionScripts $\{a_i\}_{i=0}^{n-1}$. Each PoseScript $p_i$ maps to a configuration $x_i \in \mathcal{X} \subseteq \mathbb{R}^{J \times 3}$, where $\mathcal{X}$ denotes the set of physically valid joint orientations in the SMPL space, and $J$ represents the number of joints. Each motion script $a_i \in \mathcal{A}$ specifies a transition between configurations $x_i$ and $x_{i+1}$, where $\mathcal{A}$ defines the set of valid motion primitives. The task-level planning must satisfy:

$$C_T(x_i, a_i, x_{i+1}) = 0, \quad \forall i \in \{0, \dots, n-1\} \tag{1}$$

where $C_T$ enforces a soft constraint on both physical validity and semantic consistency. Specifically, $C_T$ ensures that (1) transitions between configurations respect joint limits and maintain biomechanical feasibility, and (2) MotionScripts align with the intended behavior specified in the high-level instruction.

At the motion level, for each configuration pair $(x_i, x_{i+1})$, we compute a continuous trajectory $\tau_i : [0, 1] \rightarrow \mathcal{X}$ that satisfies the boundary conditions:

$$\tau_i(0) = x_i, \quad \tau_i(1) = x_{i+1}$$

and maintains physical constraints:

$$C_M(\tau_i(\lambda)) \leq 0, \quad \forall \lambda \in [0, 1] \tag{2}$$

where $C_M$ encompasses joint limits, collision avoidance, and dynamic feasibility constraints in simulation or real-world environments to ensure natural and physically plausible motion.

## 3 METHODOLOGY

### 3.1 OVERALL ARCHITECTURE

**Data Collection.** Our data collection and organization process, outlined in Figure 1, consists of several stages. We began by assembling a large pool of approximately 500k raw videos from a mix of sources. To clarify, our dataset does not rely entirely on uncurated internet videos; the majority ($\sim$300k videos) were curated from established academic datasets such as Kinetics, UCF101, HMDB, HM3.6M, and ActivityNet. The remaining portion ($\sim$200k videos), sampled from the People & Society category in YouTube-8M, serves to capture a broader diversity of "in-the-wild" human activities.

Videos with scores below a defined threshold were discarded. Subsequently, we employed TransNetv2 to segment each video into clips, ensuring each clip contains no scene transitions. Long sequences were split into multiple clips, resulting in a final dataset of approximately 300k clips. Dataset statistics and quality studies are provided in Supp. E.1 and E.2.

For data processing, each clip underwent Motion Estimation and Motion Description. For Motion Estimation, we applied an advanced 3D Human Pose Estimation model, *i.e.* TRAM, to extract SMPL sequences across the clip, which were then converted into PoseScript representations using predefined rules. Low-quality SMPL sequences were filtered based on metrics including physical plausibility, pose smoothness, and frame completeness. For Motion Description, we used a state-of-the-art multimodal large language model (*i.e.*, GPT-4o) to generate a behavior script for each video, structured as "[Subject] [Emotion/State/Style] [Action] [Direction/Goal] [Environment/Background]." Each clip was subsequently annotated to produce a corresponding MotionScript.

To mitigate potential noise and bias from this automated process, we also implemented a manual verification process on a subset (*i.e.*, 10k motion sequences) of the data, ensuring high fidelity in the final annotations. Dataset-specific annotating methods are provided in Supp. D.4.

**TAMP-based Framework Design.** We propose **PHYLOMAN**, illustrated in Figure 2, a hierarchical planning framework that enables interpretable control over complex behaviors while bridging the gap between linguistic instructions and physical execution. The framework implements both task-level and motion-level planning through LLMs and Diffusion-guided Control Policy. Notably, PHYLOMAN operates as a pure inference framework where individual components are trained separately through stage-wise optimization.

The task planner integrates an LLM-based behavior planner with a conditional VAE. Given a high-level instruction (e.g., "conduct an orchestra on stage"), it generates a sequence:

$$(p_0, a_0, p_1, a_1, \ldots, p_{n-1}, a_{n-1}, p_n)$$

where each PoseScript $p_i$ is transformed into its SMPL representation $x_i$ through generation. The Motion-Scripts $a_i$ define transitions between poses, with the constraint $C_T(x_i, a_i, x_{i+1}) = 0$ enforced through a combination of LLM reasoning and its learned physical priors.

The motion planner combines a motion in-betweening model with a control policy to generate trajectories. For each transition specified by $(x_i, x_{i+1})$ and $a_i$, it produces a discrete approximation:

$$\{\tau_i(k\Delta\lambda)\}_{k=0}^K, \quad \Delta\lambda = \frac{1}{K}$$

of the continuous trajectory $\tau_i(\lambda)$, while maintaining $C_M(\tau_i(\lambda)) \leq 0$. This ensures smooth, physically valid transitions that respect joint limits, balance, and collision constraints.

Finally, a control policy (e.g., MPC or RL-based methods) tracks the entire trajectory from $x_0$ to $x_n$, interpolating between waypoints using the normalized parameter $\lambda$ to ensure that the generated motion sequence maintains physical feasibility by consistently enforcing $C_M(\tau_i(\lambda)) \leq 0$ throughout execution. A sliding

window approach enables real-time performance while preserving both local stability and global task coherence. Please refer to Supp. C for more details.

## 3.2 HIGH-LEVEL BEHAVIOR PLANNING

In this section, we leverage LLMs for Task Planning, considering their unique strengths in common-sense reasoning and context understanding. This enables decomposing a complex goal description into a behavior script.

**LLM as Behavior Planner.** Building on established theories that human behavior is inherently hierarchical (Lashley et al., 1951), we decompose behaviors temporally into keyframes (discrete postures) and transitions (inter-keyframe movements). We design structural primitives that encode both keyframe postures ($p_i$) and transitional motions ($a_i$), incorporating human body priors to ensure physical consistency.

Given a natural language goal description $\mathcal{B}$, LLMs generate sequences of these primitives (*i.e.* BehaviorScript):

$$\{p_i\}_{i=0}^{n}, \{a_i\}_{i=0}^{n-1} = \text{LLM}(\mathcal{B}),$$

Through a naive Chain-of-Thoughts framework illustrated in Figure 2, we further enrich these sequences with kinematic attributes such as motion amplitude and speed. We tested both fine-tuned and zero-shot versions of LLMs of this framework in Section 4.3.

## 3.3 LOW-LEVEL MOTION CONTROL

The high-level behavior plan generated by the LLM must be translated into a physically-executable motion sequence. This low-level control process is hierarchical, comprising two critical stages: (1) the generation of a high-quality, kinematically coherent *motion prior*, and (2) the execution of this prior by a robust, *physics-based tracking policy* that ensures dynamic plausibility in a simulated environment.

**Long-term Generative Motion Prior.** A trivial approach to generating the motion prior is to first determine all keyframe poses from PoseScripts and then interpolate the transitions. This decoupled process, however, often produces unnatural movements, as the target poses are defined without considering the dynamics of the motion leading to them. To overcome this, we propose a novel model, i.e., Multi-segment Parallel Motion Diffusion Model (MP-MDM) that co-designs the transition and the target pose through a joint training scheme, yielding a more natural and coherent kinematic prior, $\mathcal{M}$.

Our model consists of two collaboratively trained components. The first is a Variational Autoencoder (VAE), adapted from PoseScript (Delmas et al., 2022), which learns a structured latent space for human poses. Its encoder $E_\phi$ maps a PoseScript $p_i$ to a latent code $z_{p,i}$, which is reconstructed into the SMPL pose $x_i$ by the decoder $D_\theta$ under a standard objective $\mathcal{L}_{\text{VAE}}$. The core of our innovation lies in reformulating the conditioning of the subsequent motion diffusion model (Ho et al., 2020; Zhang et al., 2022). Rather than being conditioned on a pre-computed target pose $x_{i+1}$, our diffusion model $\varphi_\psi$ is conditioned on the *latent representation* $z_{p,i+1}$ from the VAE's encoder. Its objective is to generate a trajectory segment $\mathcal{M}_i$ that starts at pose $x_i$ and naturally terminates in a pose consistent with the target latent $z_{p,i+1}$, while the path itself adheres to the MotionScript $a_i$. This is formulated as:

$$\mathcal{M}_i = \varphi_\psi(\mathbf{z}; x_i, \text{CLIP}(a_i), E_\phi(p_{i+1}), t).$$

To ensure these two components operate synergistically, they are trained jointly by optimizing a combined objective: $\mathcal{L}_{\text{Total}} = \mathcal{L}_{\text{Diffusion}} + \lambda \mathcal{L}_{\text{VAE}}$. This joint optimization compels the VAE to produce latent codes that serve as meaningful, diffusion-friendly targets, and the diffusion model to interpret these latents as valid trajectory endpoints. During inference, the generation process is highly parallel. First, the VAE decodes all PoseScripts simultaneously to produce the full set of keyframe poses $\{x_0, \ldots, x_n\}$. Subsequently, each transition segment $\mathcal{M}_i$ is generated in parallel, conditioned on its corresponding start pose $x_i$, motion

Table 2: Evaluation on **HumanML3D**, **GBC-10K**, and **GBC-100K** for motion generation. All baselines are trained with MotionScript and motion data only; for fairness, we truncate GBC-100K, balance GBC-10K with HumanML3D-style text, and standardize output length to 196 frames.

| Experiment | Method | R-Precision ↑ | | | FID ↓ | MM Dist ↓ | Diversity → | MultiModality ↑ |
|---|---|---|---|---|---|---|---|---|
| | | Top 1 | Top 2 | Top 3 | | | | |
| Trained & Evaluated on GBC-100K | Real | $0.501^{\pm.007}$ | $0.743^{\pm.002}$ | $0.833^{\pm.002}$ | $0.003^{\pm.001}$ | $2.426^{\pm.008}$ | $5.980^{\pm.104}$ | - |
| | MotionLCM (Dai et al., 2025) | $0.497^{\pm.003}$ | $\mathbf{0.751}^{\pm.002}$ | $\mathbf{0.827}^{\pm.005}$ | $0.816^{\pm.032}$ | $2.743^{\pm.003}$ | $3.846^{\pm.079}$ | $\mathbf{3.391}^{\pm.063}$ |
| | MDM (Tevet et al., 2022) | $0.417^{\pm.004}$ | $0.539^{\pm.003}$ | $0.629^{\pm.008}$ | $0.387^{\pm.118}$ | $2.657^{\pm.101}$ | $2.452^{\pm.162}$ | $2.207^{\pm.091}$ |
| | MotionCLR (Chen et al., 2024a) | $0.527^{\pm.003}$ | $\mathbf{0.751}^{\pm.005}$ | $0.838^{\pm.004}$ | $0.114^{\pm.000}$ | $2.472^{\pm.009}$ | $4.364^{\pm.000}$ | - |
| Trained & Evaluated on HumanML3D+GBC-10K | Real | $0.497^{\pm.004}$ | $0.663^{\pm.002}$ | $0.706^{\pm.003}$ | $0.005^{\pm.002}$ | $3.726^{\pm.006}$ | $7.266^{\pm.023}$ | - |
| | MotionLCM | $0.485^{\pm.006}$ | $0.648^{\pm.005}$ | $0.663^{\pm.007}$ | $0.641^{\pm.009}$ | $3.314^{\pm.006}$ | $7.723^{\pm.016}$ | $3.212^{\pm.082}$ |
| | MDM | $0.307^{\pm.004}$ | $0.478^{\pm.006}$ | $0.655^{\pm.005}$ | $0.296^{\pm.008}$ | $4.725^{\pm.003}$ | $7.407^{\pm.017}$ | $2.139^{\pm.082}$ |
| | MotionCLR | $0.537^{\pm.002}$ | $0.692^{\pm.006}$ | $0.761^{\pm.008}$ | $0.161^{\pm.000}$ | $3.314^{\pm.006}$ | $2.364^{\pm.000}$ | - |
| Trained & Evaluated on HumanML3D | Real | $0.511^{\pm.003}$ | $0.703^{\pm.002}$ | $0.797^{\pm.002}$ | $0.002^{\pm.002}$ | $2.794^{\pm.008}$ | $9.503^{\pm.065}$ | - |
| | MotionLCM | $0.502^{\pm.003}$ | $0.703^{\pm.003}$ | $0.805^{\pm.002}$ | $0.467^{\pm.012}$ | $2.986^{\pm.009}$ | $9.631^{\pm.065}$ | $2.172^{\pm.082}$ |
| | MDM | $0.320^{\pm.002}$ | $0.505^{\pm.004}$ | $0.607^{\pm.005}$ | $0.544^{\pm.044}$ | $\mathbf{2.452}^{\pm.162}$ | $\mathbf{9.559}^{\pm.068}$ | $2.799^{\pm.072}$ |
| | MotionCLR | $\mathbf{0.542}^{\pm.001}$ | $0.733^{\pm.002}$ | $\mathbf{0.827}^{\pm.002}$ | $\mathbf{0.099}^{\pm.003}$ | $2.981^{\pm.006}$ | $2.145^{\pm.043}$ | - |

script $a_i$, and target pose latent $z_{p,i+1}$. The final kinematic motion prior $\mathcal{M}$ is formed by assembling these concurrently generated segments. This "parallel-in-time" generation scheme ensures seamless continuity between segments and fully leverages the GPU's parallel processing capabilities, making it highly efficient for synthesizing long-term motion sequences.

**Tracking Policy for Whole-Body Control.** To translate the motion prior $M$ into physically-plausible actions, we introduce a low-level, task-agnostic Motion Tracking Policy (*e.g.*, PHC (Luo et al., 2023a) and GMT (Chen et al., 2025)). We first retarget $M$ to the humanoid robot's kinematic and DoF structure, then we train a policy through imitation learning (IL) to track these motions in a simulated environment. The objective of this policy is to execute the sequence of target poses provided by the motion prior in a closed-loop fashion, ensuring dynamic stability and physical realism.

To effectively learn from a large corpus of motion data without catastrophic forgetting, we follow the training paradigm of PHC (Luo et al., 2023a). This involves progressively training a stack of specialized primitive policies on increasingly difficult motion subsets, which are then orchestrated by a learned composer policy. The entire system is trained using Proximal Policy Optimization (PPO), guided by a reward function that incorporates an **Adversarial Motion Prior (AMP)** for naturalness. Critically, the policy's action $a_t$ specifies the target for a PD controller, and does not rely on any external stabilizing forces to preserve physical realism. See Supp. D.3 for more details of the policy.

## 4 EXPERIMENTS

**Experimental Setup** The primary objective of our experiments is to validate the hypotheses concerning the efficacy of our proposed method in generating extended human motion sequences that maintain behavioral continuity while adhering to high-level directives. Our PHYLOMAN aims to enhance the dataset's quality and granularity, yielding improved realism and detailed motion outputs. Furthermore, we conduct comparative experiments to evaluate the contributions of goal orientation, intentionality, and social dynamics within our behavior planning strategy. We conduct ablation studies to assess the impact of individual model components on overall performance.

**Evaluation Metrics.** Our comprehensive evaluation employs a suite of metrics: Multimodal Distance (MM Dist), Diversity, Success Rate (SR), Physical Error (Phys-Err), R-Precision, Fréchet Inception Distance (FID), Motion Length, and MultiModality. SR is assessed through human evaluation to gauge the practical effectiveness of generated behaviors. Detailed metric definitions and calculation methods are provided in Supp. D.2. Additionally, the details of the user study for evaluating the SR value are listed in Supp. D.5.

**Implementation Details.** We train PHYLOMAN with a batch size of 1024 over 100 epochs using the Adam optimizer with an initial learning rate of $10^{-5}$. We apply a cosine annealing schedule to decay the learning rate to $10^{-3}$. The CLIP-based similarity metric is trained on our dataset to ensure domain-specific evaluation. Notably, the CLIP model and the diffusion model are trained on different splits of our dataset to ensure unbiased evaluation. Specifically, we sampled approximately 25k motion clips to fine-tune ActionCLIP, and 2k clips to fine-tune CondMDI (Cohan et al., 2024) pre-trained on HumanML3D with T5 text encoder (Raffel et al., 2020) for fine-grained, long-horizon linguistic conditioning. Our PHYLOMAN is implemented in PyTorch, and all experiments are conducted on a single NVIDIA RTX-4090 GPU. The training time is about 6 hours per 20,000 samples, while the inference time is about 1 minute per sample with 1000 frames.

## 4.1 COMPARATIVE BENCHMARKING

In Table 2, we validate the quality of our proposed dataset by evaluating multiple state-of-the-art baseline methods (i.e., MotionCLR (Chen et al., 2024a), MDM (Tevet et al., 2022), MotionLCM (Dai et al., 2025), T2M-GPT (Zhang et al., 2023), MoMask (Guo et al., 2024) CondMDI (Cohan et al., 2024)) across three distinct configurations (GBC-100K, HumanML3D+GBC-10K, and HumanML3D). Please refer to Supp. D for more details. Since existing baselines cannot generate long sequences, we truncate GBC into shorter sequences for evaluation and employ a balanced data mixture for fair comparisons.

The experimental results demonstrate that all methods trained on GBC-100K achieve higher MultiModality scores, attributable to the fine-grained textual annotations in our dataset. This finding (1) validates the finer-grained and more semantically aligned motion descriptions in our dataset; and (2) evidences that GBC is more challenging and comprehensive than HumanML3D, closer to real-world behaviors, which substantially benefits future research. While gaps exist in FID metrics, these can be attributed to our dataset's construction from internet videos, featuring more diverse motion ranges, varied video quality, and potentially less precise text alignment.

## 4.2 LONG-SEQUENCE BEHAVIOR GENERATION

Table 3 presents our comprehensive ablation studies on long-sequence (1024 frames) motion generation, the central problem that most existing methods cannot address. Our PHYLOMAN is compared against two state-of-the-art adapted baselines (MoMask and T2M-GPT) and various ablated variants. we adopt the following components for optimal performance: CondMDI (Cohan et al., 2024) serves as the Motion Generator; Chain-of-Thought (Wei et al., 2022) functions as the LLM Planner; the text-to-pose conversion method proposed by (Delmas et al., 2022); and HOVER operates as the Controller.

Table 3: Zero-shot evaluation on GBC with PHYLOMAN using 1,000 GPT-4o BehaviorScripts. "Discard" disables a component; "Heuristic" applies rule-based template matching (Jung et al., 1994).

| Component | Methods | Phys-Err↓ | Div.↑ | Succ.↑ |
|---|---|---|---|---|
| Motion Gen. | MoMask (Guo et al., 2024) | $0.224^{\pm.028}$ | $96.3^{\pm.089}$ | $0.328^{\pm.000}$ |
| | T2M-GPT (Zhang et al., 2023) | $0.131^{\pm.094}$ | $99.9^{\pm.281}$ | $0.179^{\pm.000}$ |
| | Discard | - | - | - |
| LLM Plan. | Heuristic | $0.141^{\pm.012}$ | $19.3^{\pm.017}$ | $0.118^{\pm.000}$ |
| | Discard | $1.031^{\pm.094}$ | - | $0.067^{\pm.000}$ |
| Text-to-Pose | Heuristic | $0.101^{\pm.042}$ | $97.7^{\pm.411}$ | $0.452^{\pm.000}$ |
| | ChatPose (Feng et al., 2024b) | $0.293^{\pm.050}$ | $103.5^{\pm.239}$ | $0.613^{\pm.000}$ |
| | Discard | - | - | - |
| Controller | PHC (Luo et al., 2023a) | $0.105^{\pm.050}$ | $99.6^{\pm.447}$ | $0.793^{\pm.000}$ |
| | Discard | $0.235^{\pm.076}$ | $101.5^{\pm.253}$ | $0.762^{\pm.000}$ |
| - | **Optimal** | $\mathbf{0.093}^{\pm.039}$ | $\mathbf{109.7}^{\pm.253}$ | $\mathbf{0.821}^{\pm.000}$ |

The results unequivocally demonstrate PHYLOMAN's significant performance improvements across all key metrics: Compared to methods without TAMP, our PHYLOMAN achieves a 133% increase in the success rate (from 0.3 to 0.7) while reducing the physical error by 91% (from 1.224 to 0.105). These findings confirm that our proposed TAMP pipeline simultaneously achieves superior task completion, physical plausibility,

Table 4: Effectiveness of Hierarchical Annotations across LLM Planners. After fine-tuning on GBC-100K, PHYLOMAN shows notable gains in behavior planning, slightly surpassing closed-source LLMs.

| Setting | Model | Diversity ↑ | MultiModality ↑ | Succ. Rate ↑ |
|---|---|---|---|---|
| Fine-tuned | Llama3.1-70B (Grattafiori et al., 2024) | $105.37^{\pm.215}$ | $\mathbf{3.052}^{\pm.020}$ | $0.753^{\pm.000}$ |
| | Qwen-V2.5-72B (Yang et al., 2024) | $\mathbf{112.24}^{\pm.112}$ | $2.721^{\pm.025}$ | $\mathbf{0.821}^{\pm.000}$ |
| | DeepSeek-V3 (DeepSeek-AI, 2024) | $107.82^{\pm.419}$ | $2.629^{\pm.020}$ | $0.806^{\pm.000}$ |
| Zero-shot | Llama3.1-70B | $92.53^{\pm.226}$ | $2.224^{\pm.028}$ | $0.702^{\pm.000}$ |
| | Qwen-V2.5-72B | $96.83^{\pm.128}$ | $2.103^{\pm.030}$ | $0.708^{\pm.000}$ |
| | DeepSeek-V3 | $97.26^{\pm.722}$ | $2.157^{\pm.030}$ | $0.653^{\pm.000}$ |
| | GPT-4o (Hurst et al., 2024) | $103.84^{\pm.076}$ | $2.309^{\pm.030}$ | $0.807^{\pm.000}$ |
| | Claude-3.5-sonnet (Anthropic) | $109.21^{\pm.093}$ | $2.251^{\pm.030}$ | $0.778^{\pm.000}$ |

and naturalness: the three pillars of high-quality motion generation. For detailed case studies, please refer to Supp. D.6.

### 4.3 ANALYSIS OF PLANNING AND PHYSICAL CONSTRAINTS

**Hierarchical Planning.** We evaluate the effectiveness of our hierarchical planning framework through comprehensive quantitative and qualitative analyses. As shown in Table 3, our LLM planner significantly outperforms the Heuristic method, achieving substantially higher success rates (0.821 vs. 0.118) and greater diversity (109.7 vs. 19.32). pturing detailed motion characteristics.

**Text-to-Pose Mapping.** Compared to heuristic approaches, our PHYLOMAN exhibits a marginal decrease in Physical Error (0.093 vs. 0.101) and achieves a substantial 81.6% improvement in Success Rate (0.821 vs. 0.452) and a 12.2% enhancement in Diversity (109.7 vs. 97.73), indicating a clear advantage in the completion of practical tasks.

**Simulator.** Our comparison between variants with and without physics simulation reveals that incorporating physics reduces Physical Error by 60.5% (from 0.235 to 0.093) while maintaining identical Success Rates (0.821) and marginally increasing Diversity (by 8.0%), underscoring the importance of physical constraints in preserving motion quality.

**Fine-tuning LLMs.** Our experimental analysis shows that fine-tuning an LLM on hierarchical annotations from our dataset significantly improves motion sequence quality. As shown in Table 4, fine-tuned LLMs outperform pre-trained ones. The dataset, derived from internet videos, was annotated with motion and behavior scripts using a VLM. These annotations, encoding a structured semantic hierarchy, were used to fine-tune the LLM as a motion planner, enabling more coherent and contextually appropriate trajectories. The key insight is that internet videos inherently capture realistic human behavior patterns, providing rich semantic information. By integrating these patterns, we achieved notable performance gains, highlighting the value of real-world data and structured hierarchies in motion planning. This approach bridges raw video data with semantically rich motion generation, advancing computer vision applications.

## 5 CONCLUSION

In this paper, we presented a physics-informed framework supported by our introduced large-scale multimodal benchmark, designed for language-driven behavioral planning and physics-based motion control. This framework enables the generation of coherent, ultralong humanoid behaviors. Looking ahead, we aim to expand the framework for practical applications, including embodied intelligence and digital avatars.

## ETHICS STATEMENT

This work fully adheres to the ICLR Code of Ethics. Our study does not involve human-subjects research, the collection of personally identifiable information, or the annotation of sensitive attributes. The proposed GBC-100K dataset is constructed primarily from established academic benchmarks (e.g., Kinetics, UCF101, HMDB, Human3.6M, ActivityNet), supplemented by a non-sensitive subset of publicly available YouTube-8M videos. All data sources are strictly used under their respective licenses and terms of use. Motion annotations were generated automatically via pose estimation models and large language models to improve scalability. However, due to the inherent limitations of automated generation, these annotations were treated only as preliminary drafts. All final dataset entries were curated and verified through multiple rounds of manual screening by the authors to ensure accuracy, fidelity, and fairness. The dataset contains only non-identifiable human activity data, and its use is strictly intended for academic research in generative modeling and embodied AI.

## REPRODUCIBILITY STATEMENT

We have taken significant steps to ensure reproducibility of our results. All implementation details, including model architectures, optimization settings, evaluation metrics, and ablation protocols, are described in detail in the main text and supplementary materials. The dataset construction pipeline is documented step-by-step, and all filtering, annotation, and verification procedures are explicitly reported. To further support transparency, we commit to releasing the full codebase, pretrained models, and the GBC-100K dataset upon acceptance, enabling independent verification and extension of our work. Random seeds, hyperparameters, and computational resources (single NVIDIA RTX-4090 GPU) are also specified to facilitate faithful replication.

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

Table 5: Notations used in this paper (see also the terminology definitions in §A).

| Symbol | Description |
| --- | --- |
| $\mathcal{X}$ | Configuration space of the SMPL model |
| $\mathcal{A}$ | Action space of MotionScripts |
| $\mathcal{T}$ | Trajectory space |
| $\mathcal{J}$ | Set of all joints |
| $\mathcal{K}$ | Set of all collision pairs |
| $x_{i,j}$ | Angle of joint $j$ at configuration $x_i$ |
| $\bar{v}_j, \bar{a}_j$ | Maximum allowable velocity and acceleration of joint $j$ |
| $\sigma(c, v)$ | Smooth penalty function: $\max(0, v - c)^2$ |
| $g(a_i)$ | Feature mapping function: maps action $a_i \in \mathcal{A}$ to the feature space |
| $\bar{g}$ | Expected semantic feature vector |
| $\kappa, \epsilon_s$ | Transition steepness ($\kappa$) and semantic tolerance threshold ($\epsilon_s$) |
| $d_k(\lambda)$ | Distance between collision pair $k$ at trajectory progress $\lambda$ |
| $\mathbf{M}(x)$ | Mass matrix for SMPL |
| $\mathbf{C}(x, \dot{x})$ | Coriolis forces for SMPL |
| $\mathbf{G}(x)$ | Gravitational force vector for SMPL |
| $\tau(\lambda)$ | Joint torques at trajectory progress $\lambda$ |

## A  TERMINOLOGY AND NOTATIONS

Here, we explain the key terms and notations in our PHYLOMAN framework for readers unfamiliar with related topics.

**PoseScript.** The term "PoseScript" describes the specific configuration of the human body at a given moment in time. This configuration is expressed through the spatial characteristics of various body parts, including joint angles, inter-limb distances, relative positions, body orientations, and ground contact states.

**MotionScript.** The term "MotionScript" refers to the temporal characteristics of human movement over a period of time. It captures the process of human movement by describing aspects such as direction, amplitude, duration, and sequence of motion.

**BehaviorScript.** A "BehaviorScript" comprehensively describes a human action by integrating multiple PoseScripts and MotionScripts into a cohesive high-level expression, such as "changing a tire on a bicycle." A complete BehaviorScript consists of an abstract behavioral statement and a sequence of interleaved PoseScripts and MotionScripts, representing both static actions and their transitions. This definition enables our framework to generate transitions between actions in parallel, while still allowing each transition to be modified separately.

## B  RELATED WORK

**Behavior Decomposition.** In the field of behavior modeling, psychology and sociology have laid foundational insights by decomposing complex human actions into fundamental components, exploring the influence of cognitive processes and societal structures (Olsson et al., 2020; Jaffe et al., 2023). However, translating these theories into computational models remains challenging due to the complexity of mental states, social interactions, and environmental factors influencing human behavior (Nayebi et al., 2024). Building on foundational insights from psychology and sociology, recent advancements in video understanding and ac-

tion recognition have begun translating complex human behaviors into computational models by leveraging computer vision techniques, such as MotionLLM (Wu et al., 2024; Zhang et al., 2025a), Video-LLaVA (Lin et al., 2023a), and MiniGPT4-Video (Ataallah et al., 2024). These models capture intricate temporal dependencies and relational context in sequential actions, which is particularly beneficial for instructional video analysis (Tang et al., 2019). Here, methods that recognize both the hierarchical structure and the procedural flow of tasks have enabled a more nuanced understanding of human actions in real-world scenarios (Lan et al., 2015; Benavent-Lledo et al., 2024). Despite this progress, current methods still struggle with generating lifelike coherence and physical plausibility across continuous sequences (Lu et al., 2023; Zhang et al., 2022; He et al., 2024b). Addressing these limitations, our work aims to generate human behaviors that not only execute realistically in physical environments but also align with high-level semantic instructions, thereby advancing adaptability and realism in human behavior modeling.

**Human Motion Synthesis** encompasses several core areas: pose estimation, motion generation, motion prediction, and the application of physical constraints. In pose estimation, methods such as TRAM (Wang et al., 2025) and AiOS (Sun et al., 2024) have advanced the field by integrating techniques such as tracking and SLAM (Mur-Artal et al., 2015) to capture global trajectories and detailed human motions from in-the-wild videos. For human motion generation, generative models conditioned on inputs such as text or audio have made significant strides in creating realistic short-term movements (Ho et al., 2020; Rombach et al., 2022; Zhang et al., 2022; Lu et al., 2023). However, these models often struggle with achieving semantic coherence and physical plausibility across extended sequences. Similarly, while advances in motion prediction have refined the accuracy of forecasting future movements, these models frequently overlook physical feasibility, leading to sequences that may disrupt the coherence of generated behaviors (Zhang et al., 2024; Xie et al., 2021). The primary limitation of existing methods is the absence of a cohesive framework that can jointly handle high-level planning and enforce physical constraints over long sequences.

**Motion Control for Robotics.** Recent advancements in motion control and planning have explored various paradigms, including Task and Motion Planning (TAMP) (Garrett et al., 2021) and learning-based approaches (Zhang et al., 2015; Fu et al., 2024). TAMP integrates high-level task planning with low-level motion execution, enabling robots to perform complex tasks by considering both discrete actions and continuous movements (Chitnis et al., 2016). However, traditional TAMP methods often rely on predefined models and may struggle with adaptability in dynamic or unstructured environments (Zhao et al., 2024; Faroni et al., 2023). To address these limitations, learning-based techniques, such as reinforcement learning (RL) (Tang et al., 2024), have been integrated into TAMP frameworks, allowing robots to learn motion policies that adapt to environmental changes. Despite these advancements, RL approaches inherently struggle with generalization across diverse contexts and typically lack mechanisms for incorporating instructions, restricting their effectiveness in instruction-driven tasks and further necessitating extensive computational resources during training (Munikoti et al., 2023; Bertran et al., 2020; Kirk et al., 2023). Existing approaches to motion control aim to achieve human-like behaviors by balancing high-level task planning with detailed motion execution (Huang et al., 2022a). However, existing methods often fall short because of their inability to dynamically integrate high-level semantic goals with low-level physical feasibility across long-horizon tasks (Dulac-Arnold et al., 2021). Addressing this gap, our PHYLOMAN seeks to unify planning and control within a physics-informed framework, promoting coherent, adaptable behavior over extended sequences for the further application of embodied intelligence.

## C    BEHAVIOR CONSTRAINTS

In this section, we expound on the technical and theoretical details of our proposed approach in Section 4.1. To synthesize human motion that aligns with semantic expectations and physical feasibility, constraints are defined over three spaces: the configuration space of the SMPL (Pavlakos et al., 2019) model $\mathcal{X}$, the action space of MotionScripts (Yazdian et al., 2023) $\mathcal{A}$, and the trajectory space $\mathcal{T}$. Two key constraints are

proposed: the high-level transition constraint $C_T : \mathcal{X} \times \mathcal{A} \times \mathcal{X} \rightarrow \mathbb{R}$ and the low-level motion constraint $C_M : \mathcal{T} \rightarrow \mathbb{R}$.

## C.1 HIGH LEVEL CONSTRAINTS

The high-level constraint is defined as:

$$C_T(x_i, a_i, x_{i+1}) = w_1 f_j(x_i, x_{i+1}) + w_2 f_s(a_i),$$

where $w_1, w_2 > 0$ are weights. Here, $f_j(x_i, x_{i+1})$ enforces joint-based constraints, and $f_s(a_i)$ ensures semantic consistency.

The joint constraint $f_j(x_i, x_{i+1})$ combines range and biomechanical limits:

$$f_j(x_i, x_{i+1}) = w_a f_r(x_{i+1}) + w_b f_b(x_i, x_{i+1}),$$

where $w_a, w_b > 0$. The range constraint $f_r(x_{i+1})$ ensures each joint $j \in \mathcal{J}$ lies within anatomically plausible limits:

$$f_r(x_{i+1}) = \sum_{j \in \mathcal{J}} [\sigma(x_j^{\max}, x_{i+1,j}) + \sigma(x_{i+1,j}, x_j^{\min})],$$

where $x_j^{\min}$ and $x_j^{\max}$ represent the minimum and maximum allowable joint angles. The biomechanical constraint $f_b(x_i, x_{i+1})$ penalizes excessive joint velocities $\dot{x}_{i+1,j}$ and accelerations $\ddot{x}_{i+1,j}$:

$$f_b(x_i, x_{i+1}) = \sum_{j \in \mathcal{J}} \left[ \sigma(\bar{v}_j, \|\dot{x}_{i+1,j}\|) + \sigma(\bar{a}_j, \|\ddot{x}_{i+1,j}\|) \right],$$

and $\sigma(c, v) = \max(0, v - c)^2$ penalizes values exceeding the limits $\bar{v}_j$ (velocity) and $\bar{a}_j$ (acceleration).

The semantic function $f_s(a_i)$ aligns action $a_i$ with expected semantic features:

$$f_s(a_i) = \frac{\|g(a_i) - \bar{g}\|^2}{1 + \exp(-\kappa(\|g(a_i) - \bar{g}\| - \epsilon_s))},$$

where $g(a_i)$ is obtained by combining a diffusion model, which generates SMPL sequences, with a CLIP-based SMPL encoder (Wang et al., 2021) that extracts semantic features from these sequences. The expected semantic feature $\bar{g}$ is obtained from ground-truth SMPL sequences using the same encoder. $\kappa > 0$ controls the transition steepness, and $\epsilon_s > 0$ is the semantic tolerance.

## C.2 LOW LEVEL CONSTRAINTS

For trajectory $\tau_i$, the low-level motion constraint aggregates joint limits, collision avoidance, and dynamic feasibility:

$$C_M(\tau_i) = w_3 g_j(\tau_i) + w_4 g_c(\tau_i) + w_5 g_d(\tau_i),$$

where $w_3, w_4, w_5 > 0$ balance the terms. Practically, these constraints are derived from the Mu-JoCo (Todorov et al., 2012) simulator, ensuring realistic dynamics. In this study, MuJoCo models dynamics with discrete time steps ($\Delta t$) using semi-implicit Euler integration:

$$q_{t+\Delta t} = q_t + \Delta t \cdot v_{t+\Delta t}. \tag{3}$$

Contact forces are computed using implicit optimization methods, ensuring numerical stability during trajectory simulation. In this context, the joint path constraint $g_j(\tau_i)$ ensures limits along the trajectory:

$$g_j(\tau_i) = -\int_0^1 \sum_{j \in \mathcal{J}} \left[ \sigma(x_j^{\max}, x_j(\lambda)) + \sigma(x_j^{\min}, x_j(\lambda)) \right] d\lambda.$$

Collision avoidance $g_c(\tau_i)$ prevents violations of the minimum allowable distance $d^{\min}$:

$$g_c(\tau_i) = -\int_0^1 \sum_{k \in \mathcal{K}} \frac{\sigma(d^{\min}, d_k(\lambda))}{1 + \exp(-\kappa_c(d^{\min} - d_k(\lambda)))} d\lambda,$$

where $d_k(\lambda)$ represents the distance of the $k$-th collision pair. $\kappa_c > 0$ is the collision steepness parameter, which controls the rate at which the penalty increases as the distance $d_k(\lambda)$ approaches $d^{\min}$.

The dynamic feasibility constraints ensure that the synthesized motion trajectory adheres to the physical dynamics of the SMPL model. In Mujoco, the dynamics are governed by the equation of motion:

$$\mathbf{M}(x)\ddot{x} + \mathbf{C}(x, \dot{x}) + \mathbf{G}(x) = \tau,$$

The dynamic feasibility constraint is then formulated as:

$$g_d(\tau_i) = -\int_0^1 \|\mathbf{M}(x(\lambda))\ddot{x}(\lambda) + \mathbf{C}(x(\lambda), \dot{x}(\lambda)) +$$
$$\mathbf{G}(x(\lambda)) - \tau(\lambda)\|^2 d\lambda,$$

where $\mathbf{M}$ is the mass matrix, $\mathbf{C}$ represents Coriolis forces, $\mathbf{G}$ is gravitational force, $\tau(\lambda)$ denotes joint torques generated through control policy, and $T > 0$ is the total duration of the motion trajectory.

## D  ADDITIONAL EXPERIMENTAL DETAILS

### D.1  BASELINES

We evaluate our PHYLOMAN on a variety of baselines that achieve state-of-the-art performance in generative quality, diversity, and semantic alignment. We briefly introduce each baseline as follows:

- **T2M-GPT** (Zhang et al., 2023): Combines Vector Quantized Variational Autoencoders (VQ-VAE) with Generative Pre-trained Transformers (GPT) to produce high-quality motion sequences aligned with textual inputs.
- **MoMask** (Guo et al., 2024): Employs a generative mask modeling framework with hierarchical quantization, using masked and residual transformers to generate multi-layered high-fidelity motions.
- **MDM** (Zhang et al., 2022): Utilizes a diffusion-based generative approach, generating motions through gradual denoising guided by textual descriptions.
- **MotionLCM** (Dai et al., 2025): Learns latent representations of motion, enabling effective modeling of text-to-motion mappings in latent space.
- **MotionCLR** (Chen et al., 2024a): Applies contrastive learning to capture the correspondence between text and motion, ensuring the generated sequences align with textual inputs.
- **CondMDI** (Cohan et al., 2024): Introduces Flexible Motion In-betweening, capable of generating precise and diverse motions with flexible spatial constraints and text conditioning.

### D.2  EVALUATION METRICS

To comprehensively evaluate the proposed method, we adopt a range of metrics from MDM (Zhang et al., 2022), MotionLCM (Dai et al., 2025), and PhysDiff (Yuan et al., 2023). Each metric has been carefully selected to capture different aspects of the generated motion sequences, such as their alignment with high-level textual directives, physical plausibility, and behavioral diversity. Our evaluations leverage established methodologies from prior works, ensuring consistency and comparability with existing benchmarks.

1. **Multimodal Distance (MM Dist):** Measures the alignment between generated motions and their corresponding textual descriptions. Leveraging ActionCLIP fine-tuned on GBC-100K, we extract feature embeddings for both the generated motions and their textual counterparts. The average cosine distance between these embeddings is computed to quantify alignment, with lower values indicating better correspondence.

2. **Diversity and MultiModality:** Diversity captures the variance of generated motions across the entire dataset by calculating the pairwise feature distances between randomly sampled motion sequences, following the definitions outlined in MotionLCM (Dai et al., 2025). MultiModality, in contrast, measures the diversity of generated motions conditioned on the same textual description. This is achieved by sampling two subsets of motions for each textual description and averaging the pairwise distances between their feature embeddings. Together, these metrics reflect the richness and multi-modal nature of the generated outputs.

3. **Success Rate (SR):** Evaluates the practical utility of the generated motion sequences in completing intended tasks. To compute SR, we conducted a human evaluation study using, where participants were presented with generated motion sequences and their corresponding high-level directives. More details can be found in Supp. D.5.

4. **Physical Error (Phys-Err):** Computed following the methodology from PhysDiff (Yuan et al., 2023), it includes three components: ground penetration (Penetrate), floating violations (Float), and foot sliding (Skate). Penetrate measures the distance between the ground and the lowest mesh vertex below it, while Float measures the distance of the lowest mesh vertex above the ground. Skate quantifies the horizontal displacement of foot joints during ground contact in adjacent frames. A tolerance of 5 mm is applied to account for geometric approximations. Phys-Err is the aggregate sum of these components, providing a holistic measure of physical plausibility.

5. **Fréchet Inception Distance (FID):** Used to evaluate the quality of generated motions by comparing their feature distributions to those of ground-truth motions. FID is calculated by extracting embeddings using ActionCLIP and computing the Fréchet distance between the distributions of real and generated motions. Lower FID values indicate closer alignment between the two distributions.

6. **R-Precision:** Assesses text-motion alignment by measuring the proportion of correct matches between generated motions and ground-truth motions, given a textual description. For each description, the top-k closest motions in the embedding space are retrieved, and R-Precision is computed as the percentage of ground-truth motions among the retrieved sequences. This metric is consistent with the definitions used in MDM.

## D.3 FORMALISM OF THE POLICY

We formulate the control problem as a Markov Decision Process (MDP) defined by the tuple $\mathcal{M} = (\mathcal{S}, \mathcal{A}, \mathcal{T}, \mathcal{R}, \gamma)$. The state $s_t \in \mathcal{S}$ at timestep $t$ is composed of the robot's proprioception $s_t^p$ and a goal state $s_t^g$:

$$s_t = (s_t^p, s_t^g)$$

The proprioceptive state $s_t^p = (q_t, \dot{q}_t)$ includes the current joint configurations $q_t$ and velocities $\dot{q}_t$ of the simulated humanoid. The goal state $s_t^g$ represents the tracking objective, defined by the difference between the future reference state $(\hat{q}_{t+1}, \hat{\dot{q}}_{t+1})$ from trajectory $M$ and the current state of the humanoid, all expressed in the root-relative coordinate frame.

The policy $\pi(a_t|s_t) = \mathcal{N}(\mu(s_t), \sigma)$ outputs an action $a_t \in \mathcal{A}$, which specifies the target joint positions for a Proportional-Derivative (PD) controller. The final torque $\tau_t$ applied at each joint is calculated as:

$$\tau_t = k_p(a_t - q_t) - k_d\dot{q}_t$$

where $q_t$ and $\dot{q}_t$ are the current joint positions and velocities, and $k_p, k_d$ are the fixed controller gains. This formulation avoids the use of non-physical external forces, ensuring that the generated motions are dynamically consistent.

To produce motion that is both accurate and natural, the reward function $r_t = \mathcal{R}(s_t, \hat{q}_{t+1})$ combines multiple objectives. We adopt the reward structure from PHC, which includes a task reward for imitation $r_t^g$, a style reward from an Adversarial Motion Prior (AMP) discriminator $r_t^{\text{amp}}$, and an energy penalty $r_t^{\text{energy}}$:

$$r_t = w_g r_t^g + w_{\text{amp}} r_t^{\text{amp}} + r_t^{\text{energy}}$$

The task reward $r_t^g$ encourages the humanoid to match the reference motion across joint position, rotation, linear velocity, and angular velocity. The AMP reward $r_t^{\text{amp}}$ ensures the motion remains within the distribution of natural human movements. The energy penalty discourages high-frequency, jittery actions.

### D.4   PROMPT TEMPLATE

In our PHYLOMAN, we implement a Chain-of-Thought (CoT) approach to decompose high-level behavior instructions into structured motion sequences. This process consists of two main stages:

**Behavior Understanding and Planning** First, we employ a large language model to comprehend the high-level instruction and generate a structured behavior plan. The model outputs:

- A concise summary capturing the core behavior category
- A detailed description explaining the timing, body movements, objectives, and interactions involved

**Behavior Understanding Prompt**

You are an assistant designed to translate high-level instructions into a sequential behavior plan...

Given the following instruction: "instruction"

DO NOT output additional words or any code block, but only the summary and description. Please generate a summary and a behavior description, both in natural language. Keep the summary short with a few words.
**Example Outputs:**

```
{
    "summary": "Short summary of the instruction.",
    "description": "Behavior description for the instruction."
}
```

**Sequential Decomposition** The behavior plan is then decomposed into two complementary scripts using the following detailed prompt:

**Sequential Decomposition Prompt**

You are an assistant that transforms high-level behavior instructions into a structured, low-level, long-horizon motion sequence for single humanoids. Each element in the sequence contains both a 'keyframe' and a 'transition'. The 'transition' describes the action connecting this keyframe to the next one.

**Output Format:**

```
[
    {
        "keyframe": "Description of the first pose or state.",
        "transition": "Description of the transition to the next
    keyframe."
    },
    ...
    {
        "keyframe": "Description of the last pose or state.",
        "transition": ""
    }
]
```

**Rules for Keyframe:**

1. Identify Key Body Parts: Focus on arms, legs, head, torso
2. Use Defined Posecodes:
   - *Angle Posecodes:*
     – straight
     – slightly bent
     – partially bent
     – bent at a right angle
     – almost completely bent
     – completely bent
   - *Distance Posecodes:*
     – close
     – shoulder width apart
     – spread
     – wide apart
   - *Relative Position Posecodes:*
     – X-axis: 'at the right of', 'x-ignored', 'at the left of'
     – Y-axis: 'below', 'y-ignored', 'above'
     – Z-axis: 'behind', 'z-ignored', 'in front of'
   - *Pitch & Roll Posecodes:*
     – vertical
     – horizontal
     – pitch-roll-ignored
   - *Ground-Contact Posecodes:*
     – on the ground
     – ground-ignored
   - *Orientation Posecodes:*
     – X-axis: lying flat forward to lying flat backward

- Y-axis: leaning left to leaning right
- Z-axis: about-face turned clockwise to counterclockwise
- *Position Posecodes:*
  - X/Y/Z-axis: significant left/downward/backward to right/upward/forward

3. Subject Selection: Identify the most active joint as the subject
4. Ensure descriptions indicate static posture, not dynamic motion

**Rules for Transition:**

1. Provide the overview of human action
2. Use specified posecodes for describing changes
3. Include movement directions: Forward, Backward, Left, Right
4. Describe the speed and magnitude of movements
5. Maintain temporal relationships between concurrent movements

**Example Outputs:**
*Keyframe:*
The person is standing upright with a slight forward lean. The left arm is slightly bent and extended outward. The right arm is bent at a right angle, with the hand positioned near the chest. The legs are straight and shoulder-width apart.

*Transition:*
The person moves far to the right. At the same time, he is moving way over forward at an average pace. A moment later, he turns clockwise. The left elbow is bent at a right angle, from that pose, the left elbow is extending greatly and very fast.

This decomposition results in:

**a) PoseScript (Keyframes):** Each keyframe describes a static posture using standardized pose codes as detailed in the prompt rules.

**b) MotionScript (Transitions):** Each transition describes the dynamic motion between keyframes following the specified guidelines.

### D.5 USER STUDY

We conducted a comprehensive user study to evaluate the performance of our PHYLOMAN framework against existing motion generation baseline models. The study involved 20 participants with diverse backgrounds from our institution. Each participant was presented with 10 samples randomly selected from a sample pool containing 100 examples per method.

**Evaluation Metrics** We developed a Success Rate (Succ. Rate) metric that comprehensively evaluates the generated behaviors. This metric is normalized to [0,1] and is calculated as follows:

$$SR = w_1 C + w_2 Q \tag{4}$$

where $SR$ represents the final Success Rate, $C$ represents the completion score of necessary action steps, $Q$ represents the quality score, and $w_1 = 0.5$ and $w_2 = 0.5$ are the respective weights. The quality score $Q$ is further composed of multiple sub-metrics evaluated on a 5-point Likert scale:

- Motion fluidity ($F$)
- Body part coordination ($CO$)
- Natural rhythm ($R$)
- Transition smoothness ($S$)
- Action completeness ($AC$)
- Step completion ($SC$)
- Detail preservation ($D$)
- Text-motion alignment ($A$)

These sub-metrics are combined using the following formula:

$$Q = \frac{F + CO + R + S + AC + SC + D + A}{40},\tag{5}$$

where each component is scored from 1 to 5, with:

- Score 1: Very poor/unsatisfactory
- Score 3: Average/neutral
- Score 5: Excellent/very satisfactory

**Survey Structure.** The questionnaire was designed to evaluate each sample on all eight aspects using a standardized 5-point Likert scale. For each metric, participants were provided with clear definitions.

### D.6 CASE STUDIES

To further demonstrate the capabilities and limitations of PHYLOMAN, we present qualitative results including three successful cases and one failure case of long-horizon behavior generation in Figure 3, Figure 4, and Figure 5. The successful cases showcase: (1) Martial artist performs arts techniques, (2) Athlete training at a gym, and (3) Dancer rehearsing in a studio. These examples highlight our PHYLOMAN's ability to maintain semantic alignment and physical plausibility over long sequences, while successfully decomposing high-level behavioral goals into coherent motion sequences. In contrast, our failure case shows a scenario where a person attempts to perform swimming, where the model struggles to maintain physical balance during rapid transitions and exhibits temporal inconsistency in action sequencing. These visualizations collectively demonstrate both the strengths of our PHYLOMAN in handling structured, goal-oriented behaviors and its current limitations in extremely dynamic scenarios requiring precise physical coordination.

## E DATASET DETAILS

This section provides additional details on our proposed GBC-100K dataset, including extended statistics and a quantitative validation of our data annotation pipeline.

### E.1 DATASET STATISTICS

To illustrate the characteristics of GBC-100K, we provide a statistical comparison against existing human motion datasets in Table 6. Our dataset features trajectories with significantly larger variance, indicating a greater diversity of long-horizon movements. The lower Multi-Modal Distance (MM Dist) suggests a tighter semantic alignment between textual descriptions and motions, while the substantially higher Diversity score

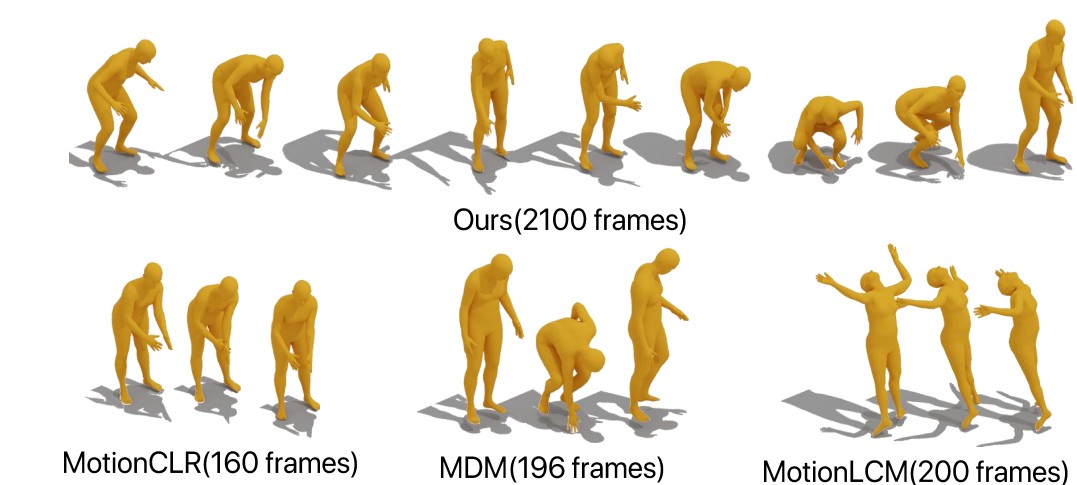

Ours(2100 frames)

MotionCLR(160 frames)    MDM(196 frames)    MotionLCM(200 frames)

Figure 3: **Qualitative Comparison of Motion Generation Methods.** Visual demonstration of motion sequences generated for the textual prompt: *"A mechanic changes a tire on a bike in a garage"*. Our proposed approach produces temporally extended sequences that better capture the complete action while maintaining semantic consistency with the textual description, outperforming baseline methods in both sequence length and motion quality.

Martial artist performs martial arts techniques.

Athlete training at a gym.

Dancer rehearsing in a studio.

Figure 4: Successful samples.

Swimmer practicing lap in a pool.

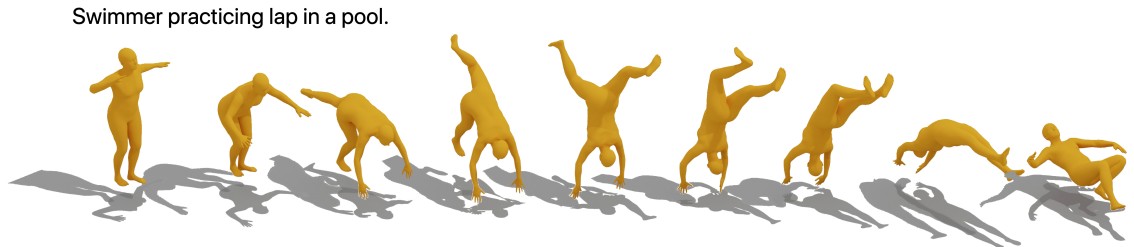

Figure 5: Failed sample.

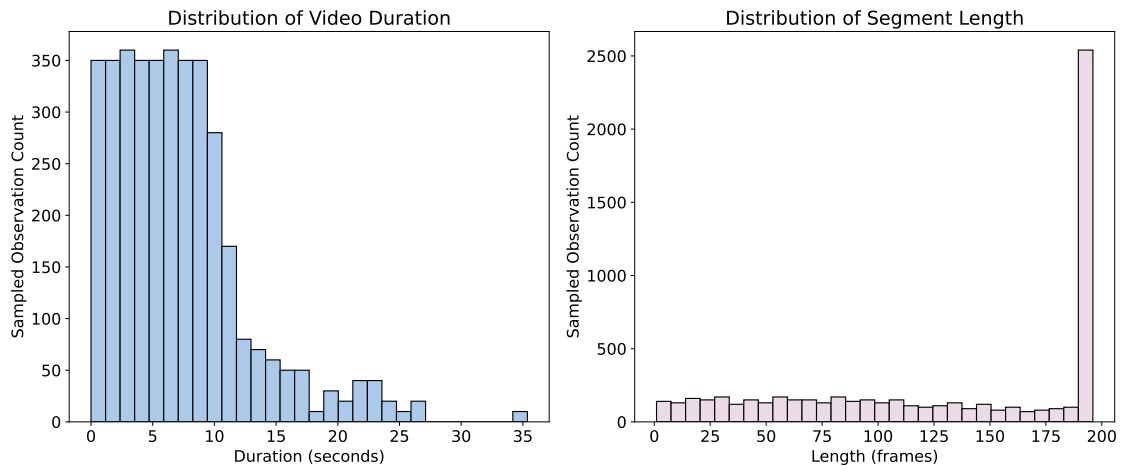

Figure 6: The duration statistics of sampled 3780 video clips and corresponding annotated motion segments with MotionScripts.

highlights the broad range of behaviors captured. Figure 8 further visualizes the rich semantic space of our dataset's textual annotations compared to others.

Table 6: **Data distribution and quality comparison.** We show the mean and standard deviation of full-length trajectories and joint positions across different datasets. We also report CLIP-based semantic similarity (MM Dist) and Diversity.

| Dataset | Joint | | Traj. | | MM Dist ↓ | Diversity ↑ |
|---------|-------|-----|-------|-----|-----------|-------------|
| | Mean | Std | Mean | Std | | |
| HumanML | 0.385 | 0.210 | 0.105 | 0.699 | 2.750 | 9.503 |
| MotionX | 0.372 | 0.198 | 0.065 | 1.187 | 2.476 | 13.174 |
| **GBC-100K** | **0.368** | **0.275** | **0.021** | **2.615** | **2.318** | **99.467** |

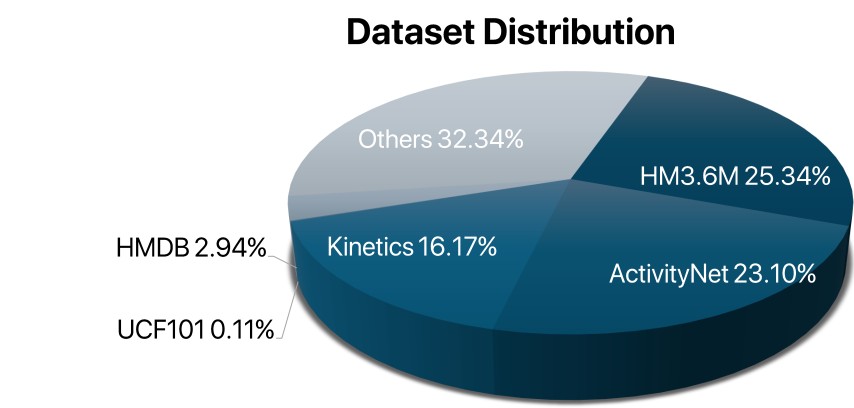

Figure 7: **Data composition in GBC-100k.** The dataset comprises contributions from multiple sources: HM3.6M (25.34%), ActivityNet (23.10%), Kinetics (16.17%), HMDB (2.94%), UCF101 (0.11%), and others (32.34%). The "Others" category includes curated subsets from the Motion-X (Lin et al., 2023b) and FLAG3D (Tang et al., 2023) motion capture datasets, as well as selected videos from YouTube-8M.

As shown in Figure 9, Figure 7, and Figure 6, our GBC-100k dataset is a large-scale, multimodal resource designed to support research on generative behavior control. It consists of diverse video-SMPL-text triplets, where each sample includes a video clip, its corresponding SMPL pose sequence, and textual descriptions in the form of BehaviorScripts, MotionScripts, and PoseScripts. The dataset integrates data from multiple well-known sources, including HM3.6M, ActivityNet, Kinetics, HMDB, UCF101, and curated sources such as Motion-X and FLAG3D, alongside selected videos from YouTube-8M, which collectively account for 32.34%. This broad composition ensures a wide range of human activities, providing diversity in both motion and context. Each source contributes to the richness of the dataset, making it a benchmark for tasks requiring fine-grained motion understanding and behavioral annotation.

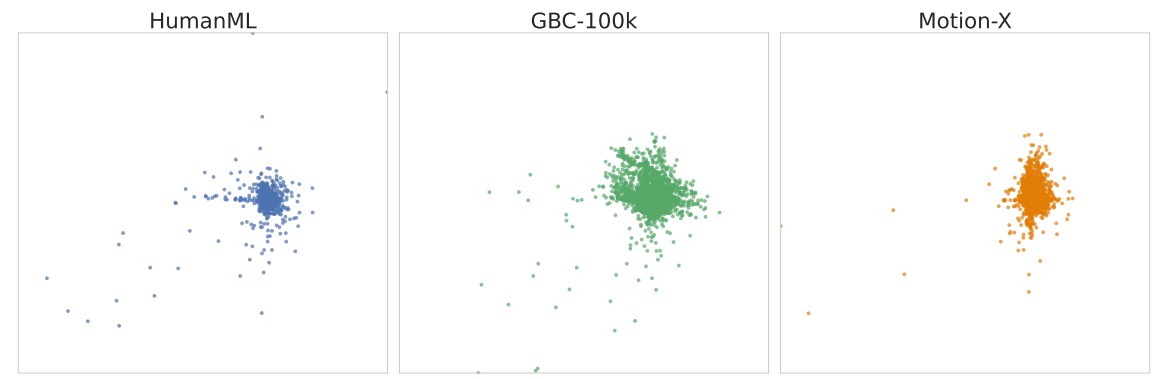

Figure 8: **t-SNE visualization of textual annotations.** Textual annotations from GBC-100K (right) cover a broader and more diverse semantic space compared to HumanML3D (left) and Motion-X (middle), as visualized by t-SNE embeddings from all-MiniLM-L6-v2.

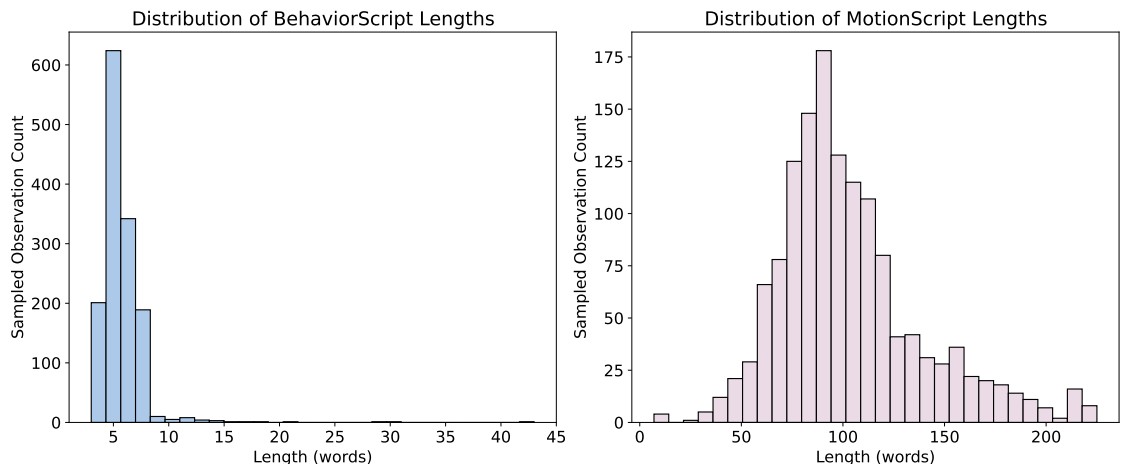

Figure 9: The Length statistics of sampled 1000 BehaviorScripts and corresponding MotionScripts.

The dataset features detailed statistics that emphasize its comprehensive scope. The sampled video clips predominantly range from 5 to 10 seconds in duration, ensuring that human actions are captured with sufficient temporal resolution. Annotated textual descriptions, including BehaviorScripts, average 10 to 15 words in length, offering concise yet informative summaries of the actions and context within the videos. MotionScripts, derived from these high-level descriptions, align closely with SMPL pose sequences, with segment lengths typically spanning 100–200 frames. Additionally, the dataset showcases linguistic diversity, as highlighted in the BehaviorScript word cloud, reflecting a wide array of actions, environments, and behavioral contexts. These characteristics make GBC-100k a versatile and robust dataset for advancing research in behavior modeling and multimodal learning.

### E.2 DATA ANNOTATION QUALITY

We provide a quantitative evaluation of our data annotation pipeline on the EMDB 2 benchmark. As shown in Table 7, our pipeline achieves state-of-the-art or competitive performance across multiple metrics, confirming its reliability for constructing the GBC-100K dataset. While weakly supervised captioning can introduce noise, we have implemented a manual verification process on a curated subset (*i.e.*, 10k motion sequences) of the data and will continue to refine the annotations for the public release. During this verification, we identified several typical error types: (1) **Motion Estimation Artifacts**, such as physically implausible poses, foot-skating, or temporal jittering in the extracted SMPL sequences; and (2) **Text-Motion Misalignment**, where the generated textual descriptions were either too generic (e.g., "a person moves" for a complex dance sequence), factually incorrect (e.g., misidentifying the active limb), or failed to capture the primary intent of the action. These findings are guiding our ongoing efforts to improve data quality.

Table 7: **Evaluation of our data annotation pipeline.** We report performance on the EMDB 2 benchmark. RTE is in %, and other pose metrics are in $mm$. Our pipeline demonstrates strong performance, ensuring high-quality motion data.

| Models | EMDB 2 | | | |
|---|---|---|---|---|
| | PA-MPJPE | WA-MPJPE$_{100}$ | W-MPJPE$_{100}$ | RTE |
| TRACE | 58.0 | 529.0 | 1702.3 | 17.7 |
| GLAMR | 56.0 | 280.8 | 726.6 | 11.4 |
| SLAHMR | 61.5 | 326.9 | 776.1 | 10.2 |
| WHAM (w/ DROID) | 38.2 | 133.3 | 343.9 | 4.6 |
| **Ours** | **38.1** | **76.4** | **222.4** | **1.4** |

## STATEMENT ON THE USE OF AI ASSISTANCE

The entirety of this manuscript, including conception, methodology, experiments, and analysis, was developed by the authors. A Large Language Model (LLM) was employed only in two limited ways: (1) as a language assistant to check grammar and improve readability, and (2) as part of the data annotation pipeline to generate motion descriptions. Since the raw outputs of LLMs are inherently limited and may contain noise, these annotations were treated only as auxiliary signals. All final dataset entries were curated through multiple rounds of manual screening and verification by the authors, ensuring fidelity, fairness, and compliance. The LLM was not involved in generating scientific content, designing experiments, analyzing data, or drawing conclusions. All intellectual contributions and research insights are solely attributable to the authors.

