# OpenReview forum: "From Motion to Behavior: Hierarchical Modeling of Humanoid Generative Behavior Control"
_ICLR.cc/2026/Conference — ICLR 2026 Conference Withdrawn Submission_

### Official Review · Reviewer_4Xj3 · 2025-10-15

**Soundness:** 2
**Presentation:** 3
**Contribution:** 2
**Rating:** 4
**Confidence:** 5

**Summary:**

This paper introduces Generative Behavior Control (GBC), a new task aimed at synthesizing long-horizon, goal-directed, and physically plausible humanoid behaviors. The authors present two main contributions:

* PHYLOMAN framework, which integrates LLM-based planning, a novel Multi-segment Parallel Motion Diffusion Model (MP-MDM), and physics-based control for behavior generation.

* GBC-100K dataset, a large-scale dataset combining SMPL motion estimations and hierarchical textual annotations, designed to support and evaluate GBC.

Experiments across HumanML3D and GBC-100K are reported, with claims that the method generates more diverse, semantically coherent, and longer motion sequences than prior baselines

**Strengths:**

1. Ambitious scope: The work reframes the field from motion generation to behavior generation, highlighting the importance of goal-directedness.

2. Scalability: Dataset construction is large-scale, leveraging ∼500k videos and semi-automated annotation pipelines, which could benefit the community if released.

3. Long-horizon motion generation: The MP-MDM parallel generation strategy is technically interesting and addresses efficiency for multi-second or minute-long behaviors.

**Weaknesses:**

1. Dataset reliability:

* The dataset relies on monocular SMPL estimation (TRAM) as its “gold standard,” which is problematic because SMPL often drifts in translation even when subjects are static (e.g., the provided example clip (`H--TB3aFpxY_000115_000125`) shows the person standing still while SMPL translation varies). This undermines claims of physical plausibility.

* Using noisy pseudo-ground-truth motion as the foundation of a benchmark (evaluation target) introduces significant bias; thus, it is questionable whether GBC-100K can be considered a reliable benchmark.

2. Evaluation design flaws:

* In Table 2, comparisons confound dataset quality with model design since training and test data are simultaneously altered. To properly assess the dataset’s contribution, training sets should vary while the test set remains fixed.

* In Tables 3 and 4, if the evaluation test set is indeed GBC-100K, then performance comparisons may simply reflect train-test overlap. Distribution similarity between train and test splits risks inflating results and obscures real generalization ability.

* Physical grounding gap: Despite emphasizing “physics-informed” behavior generation, much of the evaluation remains in SMPL parameter space without showing how the motions transfer to physically simulated humanoids. GBC-100K is based on monocular SMPL estimations, which often have severe physics artifacts.

**Questions:**

Please see Weaknssses.

---

### Official Review · Reviewer_NKyX · 2025-10-30

**Soundness:** 2
**Presentation:** 3
**Contribution:** 2
**Rating:** 4
**Confidence:** 2

**Summary:**

This paper introduces Generative Behavior Control (GBC), a new task for generating long-term, goal-oriented, and physically plausible humanoid behaviors. The authors propose PHYLOMAN, a hierarchical framework combining LLM-based high-level planning and physics-informed motion control, supported by the new GBC-100K dataset. Experiments show improvements in behavior diversity, semantic alignment, and motion length compared to baseline methods.

**Strengths:**

- GBC formalizes long-term behavior generation, addressing key gaps in motion generation research.
- PHYLOMAN integrates hierarchical planning and physics-based control, bridging high-level semantics and low-level execution.
- GBC-100K provides a valuable, hierarchically annotated benchmark for behavior generation.

**Weaknesses:**

- Claims of goal-orientation and semantic coherence lack rigorous task-driven evaluation.
- Comparisons are primarily with motion generation methods, not task-and-motion planning approaches.
- Automated annotations may introduce noise; dataset limitations are not fully analyzed.
- Lack of detailed ablations to isolate contributions of hierarchical planning and MP-MDM.

**Questions:**

Please see Weaknesses for details.

---

### Official Review · Reviewer_1oWQ · 2025-11-01

**Soundness:** 3
**Presentation:** 3
**Contribution:** 2
**Rating:** 4
**Confidence:** 4

**Summary:**

This paper introduces a hierarchical framework, PHYLOMAN, for Generative Behavior Control (GBC), combining language-driven planning, diffusion-based motion generation, and physics-based control, and constructs a large-scale hierarchical text-to-motion dataset.

**Strengths:**

1.This paper introduces a hierarchical framework, PHYLOMAN, for Generative Behavior Control (GBC), combining language-driven planning, diffusion-based motion generation, and physics-based control.
2.The paper constructs a large-scale hierarchical text-to-motion dataset with three levels of structured annotations: BehaviorScript, PoseScript, and MotionScript.

**Weaknesses:**

1.While the proposed PHYLOMAN framework is structurally coherent, its components—an LLM-based planner, a motion diffusion model, and a physics controller—are largely based on existing paradigms.
2.Although the paper cites MotionAgent (Wu et al., 2024) as a representative language-to-motion framework, there is no direct experimental comparison and analysis.
3.In the main experimental section, PHYLOMAN is not included in the key comparison Table 2, which presents quantitative results across baselines. The authors should include PHYLOMAN in Table 2 using the same configuration and evaluation metrics as other methods.
4.The GBC-100K dataset, described as containing 123.7K motion sequences and 250 hours of video, introduces hierarchical annotations: BehaviorScript, PoseScript, and MotionScript. While this is valuable, several issues arise:
The reported W-MPJPE ≈ 222 mm (Table 7) remains quite large for a high-quality motion dataset. The evaluation only includes PA-MPJPE, W-MPJPE, and RTE,and while the authors acknowledge the presence of typical error types in their data, it is necessary to address additional aspects of physical consistency, such as foot sliding, body penetration, failure ratio, and temporal jitter.
There is no analysis of long-horizon temporal consistency, which is crucial for “ultra-long” behaviors.
5. The paper repeatedly refers to the motion diffusion backbone as “parallel-in-time”, implying computational efficiency. However, there is no quantitative evidence (e.g., speedup, training cost, or memory footprint) to support this claim.

**Questions:**

Please refer the Weakness.

---

### Official Review · Reviewer_iC6n · 2025-11-03

**Soundness:** 3
**Presentation:** 3
**Contribution:** 2
**Rating:** 4
**Confidence:** 4

**Summary:**

The paper proposes a hierarchical planning and diffusion-based framework (PHYLOMAN), and builds a new dataset GBC-100K for long-term behavior generation. The motivation—bridging semantic intention and physical motion—is relevant and aligned with the community’s long-term goals.

**Strengths:**

- The paper highlights a meaningful research gap, moving from short-term motion generation to long-horizon, goal-directed behavior control, which is conceptually valuable for embodied AI.

-The paper proposes a full pipeline combining language planning, motion generation, and physics-based execution.

-The proposed dataset GBC-100K is relatively large compared to many prior datasets and includes hierarchical semantic annotations, which could support richer planning and evaluation of long-duration behaviors.

**Weaknesses:**

- The proposed framework mainly combines existing components: LLM-based behavior planning, Diffusion motion models,  and Physics-based controllers. The integration appears incremental without introducing new theoretical insights or algorithmic advances. Claiming a “first unified solution” is overstated, given recent works combining language, motion priors, and controllers.

-The dataset is largely auto-annotated using pose estimation with LLM captioning, raising concerns about noise and annotation quality.

-The structure of PoseScript with MotionScript is a data-level decomposition; the LLM planner is not grounded in physically valid constraints during generation, contradicting the claimed TAMP formulation.

-The extremely long-horizon results rely heavily on truncation and indirect evaluation metrics. Physical plausibility and task completion demonstrations are limited and lack real-world testing or robotics integration.

**Questions:**

-Since the majority of annotations come from automated VLM captioning and pose estimation, how to evaluate the annotation quality? And how sensitive is the model performance to annotation errors?

-The LLM planner is said to enforce physical feasibility (CT). How is this concretely implemented? Is there any explicit consideration in the model to ensure transitions are executable before feeding to the diffusion model?

-How is “success” defined for behaviors with abstract semantics, e.g., dancing happily?

-Are baselines given text annotations consistent with their original training domains?

-Does the system degrade gracefully with longer horizons, e.g., multi-minute outputs?

---

### Note · Authors · 2025-11-12

I have read and agree with the venue's withdrawal policy on behalf of myself and my co-authors.